# Perennial Ryegrass (*Lolium perenne* L.) Response to Different Forms of Sulfur Fertilizers

**Grzegorz Kulczycki** [1],*[iD], **Elżbieta Sacała** [1][iD], **Anna Koszelnik-Leszek** [2] **and Łukasz Milo** [3]

1 Institute of Soil Science Plant Nutrition and Environmental Protection, Wrocław University of Environmental and Life Sciences, Grunwaldzka Str. 53, 50-363 Wrocław, Poland; elzbieta.sacala@upwr.edu.pl

2 Department of Botany and Plant Ecology, Wrocław University of Environmental and Life Sciences, Grunwaldzki Sq. 24A, 50-363 Wrocław, Poland; anna.koszelnik-leszek@upwr.edu.pl

3 Chemical Plants "Siarkopol" TARNOBRZEG Ltd., Chemiczna Str. 3, 39-400 Tarnobrzeg, Poland; lmilo@zchsiarkopol.pl

* Correspondence: grzegorz.kulczycki@upwr.edu.pl; Tel.: +48-71-320-5654

**Abstract:** The aim of the study was to compare sulfate fertilizers and mixtures of elemental sulfur ($S^0$) and sulfate in terms of yield and nitrogen (N) and sulfur (S) status in perennial ryegrass. Mixtures of sulfate and $S^0$ can reduce the consumption of sulfate alone. The plants were grown in soil cultures. The plants were supplemented with $S^0$, $K_2SO_4$, $MgSO_4$, and $(NH_4)SO_4$ or a mixture of these salts with $S^o$. Two sulfur doses were applied and the ryegrass was harvested three times. Fresh and dry weights of each swath, the N and S content, and their uptake were determined. The total fresh yield of sulfur-fertilized plants was 25 to 94% higher compared to unfertilized plants. The increases in dry matter were even more significant. Fertilizers, being a mixture of $S^0$ and sulfate, showed the same efficiency as those containing sulfate alone. Sulfur fertilization resulted in a higher S content and its uptake, lowered N concentration in second and third swatch, and a decrease in total N uptake. In conclusion, to achieve high crop yields, soil sulfur deficiency should be corrected and fertilizers that are the mixture of elemental sulfur and sulfate are a beneficial and effective approach.

**Keywords:** elemental sulfur; sulfate; N:S ratio; ryegrass





## 1. Introduction

Sulfur (S), a macronutrient found in plants in the smallest amounts relative to other essential macronutrients, is currently recognized as one of the most important yield-forming elements [1,2]. In recent years, a shortage of plant-available sulfur in soils almost all over the world has been [3–6]. This adverse phenomenon is due to a drastic reduction in air pollution by sulfur, and consequently, a reduction in its deposition in the soil [6,7]. High-yielding crops grown today take up large amounts of sulfur from the soil and exacerbate this problem. It is expected that this phenomenon will intensify in the coming years [6,8] and in consequence, many agricultural areas will require fertilization with sulfur to maintain high yields and quality.

Therefore, research concerning sulfur application in agricultural plant production that will indicate an environmentally favorable, cheap for farmers, and efficient for plants solution of this problem is an urgent need. For proper growth and development, plants require sulfur at a level of 0.1–1.0% on a dry weight basis, and the average concentration of S in plant tissues ranges from 0.2 to 0.5% [9]. The main inorganic form of S directly available to plants is sulfate ($SO_4^{2-}$), which in mineral fertilizers, occurs as calcium, magnesium, potassium, or ammonium sulfate. Unfortunately, sulfate can be easily leached from the root zone, particularly from sandy soil. Furthermore, under sulfur limitation, plants absorb $SO_4^-$ very rapidly, which results in the formation of a sulfate depletion zone. On the other hand, sulfur is classified as an immobile mineral element in plants, which is not readily remobilized to younger leaves during deficiency. Its deficiency at any stage of plant

growth negatively affects plant metabolism and growth, and ultimately results in reduced yields [10]. This is due to the fact that sulfur is a multifunctional element which is not merely a component of certain amino acids (cysteine, methionine) and consequently proteins, but also builds other important cellular compounds [9,11]. They include coenzymes, lipids, and secondary metabolites that are involved in cellular metabolism, plant reaction to environmental stresses, and interactions with animals and pathogens [12]. Protein synthesis requires an adequate amount of two key macro-elements, nitrogen and sulfur; hence, a correct balance of these nutrients is particularly important [13].

Literature data indicate that sulfur and nitrogen metabolism are linked, and sulfur availability improves nitrogen uptake by plants, thereby affecting their quality and optimizing the N:S ratio [14,15]. The N:S ratio is considered to be a responsive indicator of sulfur supply to plants [16–19]. Fertilization with sulfur increases the total content of sulfur and sulfates in plants and deficit of this element results in the accumulation of non-protein nitrogen compounds that reduce the biological value of crop plants [15,20].

Fertilizers that are a mixture of sulfate and elemental sulfur ($S^0$) can be a good way to optimize sulfur supply to plants and improve its bioavailability in the long term. At the same time, they are cheaper and more environmentally friendly than sulfates alone. Elemental sulfur is chemically inert, difficult to leach from the soil compared to the anionic sulfate form, and available for longer in the soil. $S^0$ is a suitable source of this element for plants but must first be oxidized to the sulfate form by soil microorganisms. Oxidation of elemental sulfur and mineralization of organic sulfur, and thus the amount of available sulfur depends on the microbial activity of the soil. The temperature and moisture of the soil and its physico-chemical properties have a significant impact on this activity [11]. Sulfate is actively taken up through the roots and translocated to the shoot, and can be stored in vacuoles [9]. Plants are able to take up sulfate from the soil over a wide range of concentrations through the use of high-affinity and low-affinity transporters [11].

Literature data indicate that the application of sulfur increases the yield of crop and pasture plants and the magnitude of the response is dependent on the type of plant [21–23]. Some crops, such as oilseed rape and mustard, respond very well, while others respond much less well. There is little information on the response of meadow grasses; however, it is known that multi-cut grasses are more susceptible to sulfur deficiency than other crops [24]. In general, sulfur fertilization is expected to increase yields by an average of 25% under conditions of severe sulfur deficiency [25,26].

Perennial ryegrass—a species of the grass family—grows wild almost throughout Europe, northern Africa, the temperate zone of Asia, and was artificially introduced in North America and Australia. It is an excellent pasture grass that forms the basis of productive pastures for cattle. It can also be used to stabilize soil and prevent soil erosion, as well as to create hardy turf for lawns and golf courses. It has produced many varieties. Ryegrass is a highly productive plant and can be harvested multiple times during the growing season.

There is a relatively small number of studies that examine the sulfur fertilization needs of grasslands, especially in the temperate zone and over the past decade under conditions of very limited sulfur input from the atmosphere. Aspel et al. [27] in lysimeter experiments on perennial ryegrass showed that sulfur fertilization increased crop yields, nitrogen recovery from fertilizers, and significantly reduced nitrate leaching.

The aim of the study was to compare sulfate fertilizers and a mixture of elemental sulfur ($S^0$) and sulfate in terms of yield and N and S status in perennial ryegrass.

## 2. Materials and Methods

### 2.1. Pot Experiment

Plants were grown in pots filled with 2.5 kg of a sandy Arenosol whose granulometric composition was as follows: sand 86% with dominant medium and fine fractions, silt 12%, and clay 2%. The total content of carbon determined by the Dumas dry combustion method [28] and total content of sulfur determined by the Butters–Chenery method [29]

were very low (Table 1). The plant-available phosphorus, potassium, magnesium, and sulfate sulfur form content in the soil are given in Table 1. The soil had a very acidic pH prior to liming, a high content of phosphorus, and a low content of potassium and magnesium. Before sowing, the soil acidity was adjusted to pHKCl 6.62 by applying lime at a rate $CaCO_3$ (2.12 g kg$^{-1}$). The level of total sulfur content determined in the soil, along with S-SO$_4$, classifies this soil type as low sulfur soil. Macronutrients were applied to the soil before sowing in the following doses: N 104 mg kg$^{-1}$, P 124 mg kg$^{-1}$, K 293 mg kg$^{-1}$, and Mg 92 mg kg$^{-1}$. The amounts of applied doses of macronutrients (N, P, K, Mg) were balanced taking into account the amount added with examined sulfates so that they were even and the same in all pots.

**Table 1.** Physical-chemical soil properties before the experiment.

| Agronomic Category/Soil Texture | pH | C total | N total | S total | P | K | Mg | S-SO$_4$ |
|---|---|---|---|---|---|---|---|---|
| | | | | | Soluble Forms | | | |
| | 1M KCl dm$^{-3}$ | g kg$^{-1}$ soil | | mg kg$^{-1}$ | mg kg$^{-1}$ Soil | | | |
| Light/Loamy sand | 3.9 | 6.18 | 0.57 | 115 | 74 | 78 | 13 | 6.82 |

### 2.2. Schedule of Experience

Two sulfur doses of 60 and 120 mg S kg$^{-1}$ soil were applied in the form of elemental sulfur (S$^0$), sulfates (K$_2$SO$_4$, MgSO$_4$, (NH$_4$)$_2$SO$_4$), and a mixture of elemental sulfur and sulfates. The following treatments were established: no S—control soil without sulfur fertilization; S$^0$—elemental sulfur application in the form of Wigor S fertilizer ("Siarkopol" Tarnobrzeg Co. Ltd., Poland); S$^0$ + K$_2$SO$_4$; K$_2$SO$_4$ alone; S$^0$ + MgSO$_4$; MgSO$_4$; S$^0$ + (NH$_4$)$_2$SO$_4$; (NH$_4$)$_2$SO$_4$). Each treatment included four repetitions (pots).

### 2.3. Cultivated Plant and Its Vegetation

In the pot experiment, perennial ryegrass (*Lolium perenne* L. variety Solen) was cultivated. About 100 seeds (0.3 g) of perennial ryegrass were sown in 20 holes and covered with ~0.5 cm of soil. Soil moisture was maintained at 60% water-holding capacity by adding deionized water. Plants were grown in a growth chamber under controlled conditions (photoperiod 16 h/8 h light/dark and temperature 26–28 °C/16–18 °C day/night). The grass was cut after 30, 45, and 60 days of vegetation.

### 2.4. Sample Preparation and Methods for Chemical Analysis

Representative soil samples were taken before and after vegetation experiments. Soil pH was determined by the potentiometric method using 1 mol dm$^{-3}$ KCl, the total C and N content by analyzer (LECO) [28], total S content by the Butters–Chenery method [29], sulfate(VI) content by the Bardsley and Lancaster method [30], soluble forms of phosphorus and potassium by the Egner–Riehm method [31] and magnesium by the method in [32].

Plants were collected after each grass cutting and the fresh mass of ryegrass was determined. Then, plants were dried at 105 °C for one hour (to kill plant tissues and avoid dry matter loss through respiration) and then at 60 °C to constant weight. Total nitrogen level was determined by the combustion method [28], and total S content by the Butters–Chenery method [29].

### 2.5. Statistical Analysis

All results obtained were subjected to one-way analysis of variance. Prior to performing the analysis of variance, a test for homogeneity of variance within groups was performed using Levene's test and the Shapiro–Wilk test of variables' conformity to normal distribution. The significance of differences between the averages was assessed using the Tukey's post hoc test with a significance level of $p < 0.05$. For all statistical analyses, the statistical program R was used [33].

## 3. Results and Discussion

### 3.1. Fresh and Dry Mass

Under the influence of 60 mg sulfur, the highest fresh and dry total mass was found for ammonium sulfate, and the observed increases were 94 and 139%, respectively, compared to the control (without S fertilization) (Figure 1). Similar increases were observed for the mixtures of elemental sulfur and sulfate ($S^0$ + $K_2SO_4$, $S^0$ + $MgSO_4$). Remaining S fertilizers also resulted in a statistically significant stimulation of ryegrass growth but not so spectacular. Elemental sulfur alone increased the total fresh and dry mass of ryegrass by 25 and 33%, respectively.

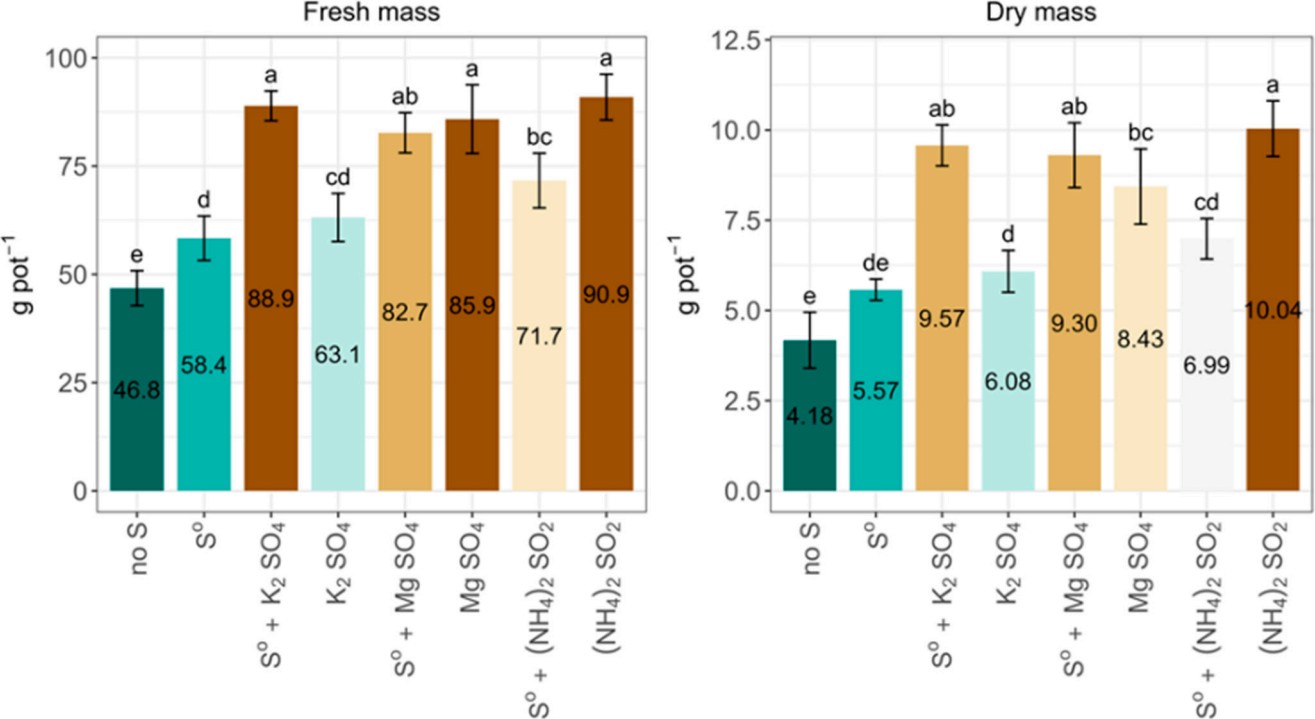

**Figure 1.** Total fresh and dry mass of ryegrass depending on sulfur fertilizer at a rate of S 60 mg/kg soil. Values labeled with the same letter are not significantly different ($p < 0.05$).

The response of ryegrass to the higher dose of sulfur (120 mg) was similar to the response to the lower dose and the least effective fertilizers were elemental sulfur and a combination of $S^0$ and $(NH_4)_2SO_4$ (Figure 2). These results show that ryegrass reacts very well to sulfur fertilization and a mixture of $S^0$ and sulfate gives significantly better results than the use of elemental sulfur alone and comparable results with the use of sulfates alone.

Figure 3 shows the dry matter yield of ryegrass obtained on the three harvest dates depending on the form of sulfur fertilizer used and the dosage rate of this element. In all the terms of ryegrass harvest, the dry matter yield obtained on fertilizers containing $S^0$ + S-$SO_4$ was similar to the objects where only sulfate forms were fertilized. The exception was the object with $S^0$ + $(NH_4)_2SO_4$ at the first harvest date, where significantly lower yields of ryegrass were obtained. As the vegetation period lengthened, the differences in ryegrass dry matter yields between the fertilizers applied decreased.

In order to visualize the differences between the fertilizers used, they were grouped according to the form of sulfur they contained (Figure 4). In all harvest dates, the combination of the $S^0$ + S-$SO_4$ form allowed dry matter yields to be obtained that did not differ from those obtained when fertilizing with the sulfate form alone.

Dry matter yields of ryegrass were also compared in relation to the sulfur doses applied (Figure 5). It was found that there were no significant differences between the

sulfur doses used, either in the total dry matter yield or in the yields obtained at different harvest periods.

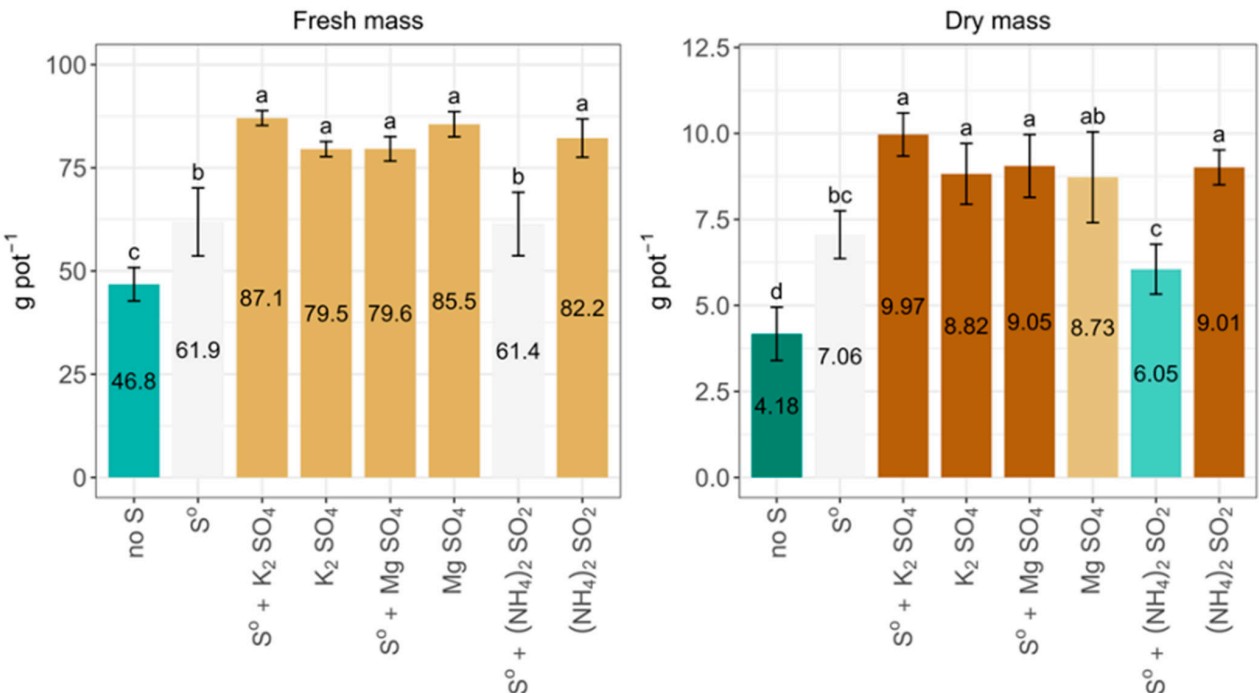

**Figure 2.** Total fresh and dry mass of ryegrass depending on S fertilizer at a rate of S 120 mg/kg soil. Values labeled with the same letter are not significantly different ($p < 0.05$).

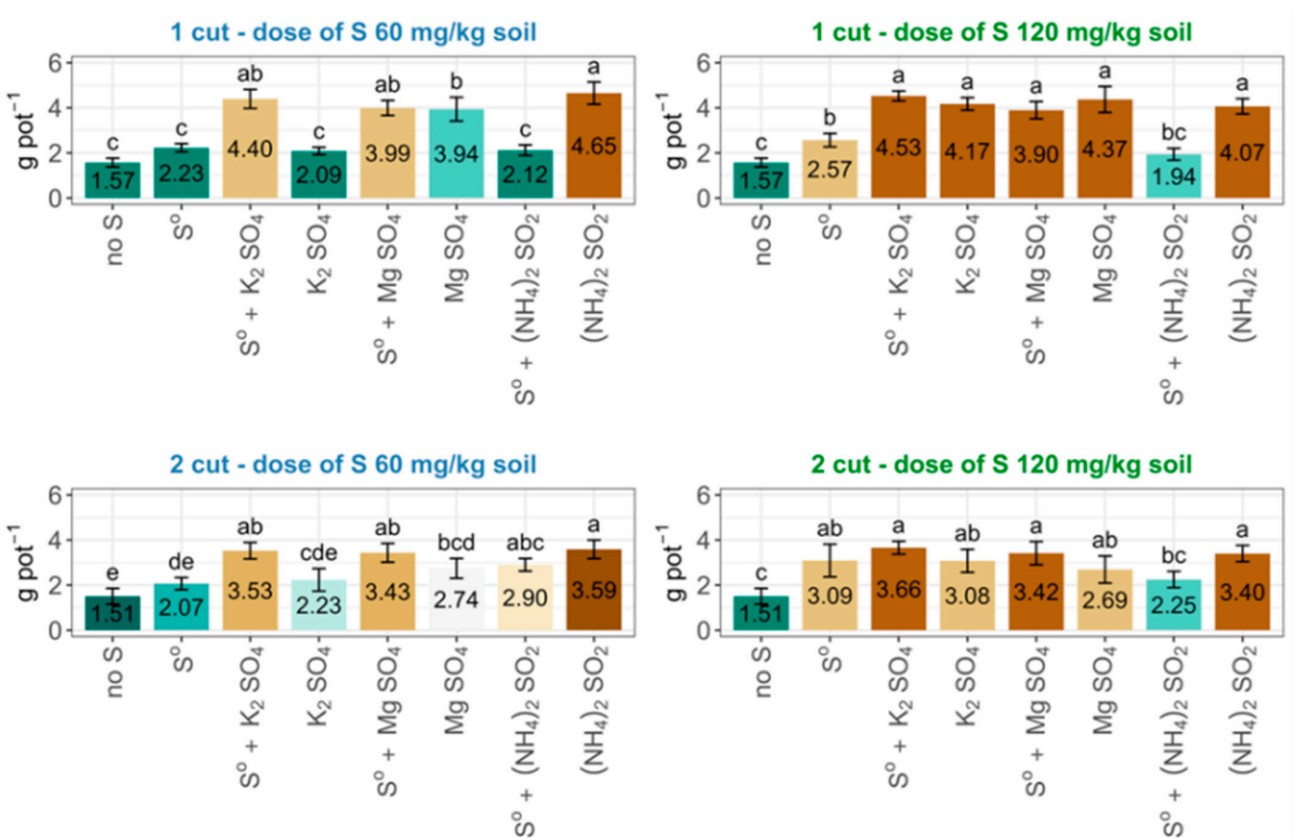

**Figure 3.** *Cont.*

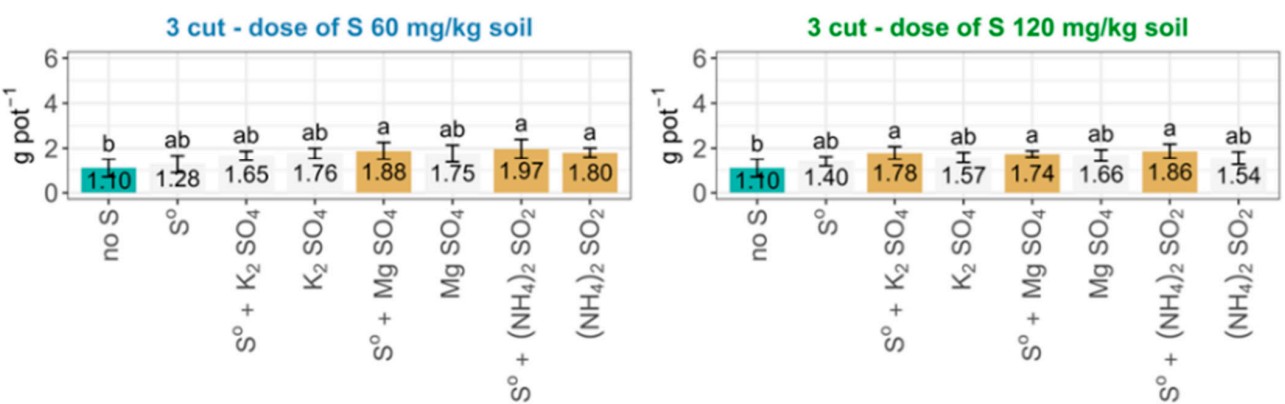

**Figure 3.** Dry mass of ryegrass depending on S fertilizer. Values labeled with the same letter are not significantly different (*p* < 0.05).

**Figure 4.** Comparison of dry mass of ryegrass according to forms of S. Values labeled with the same letter are not significantly different (*p* < 0.05).

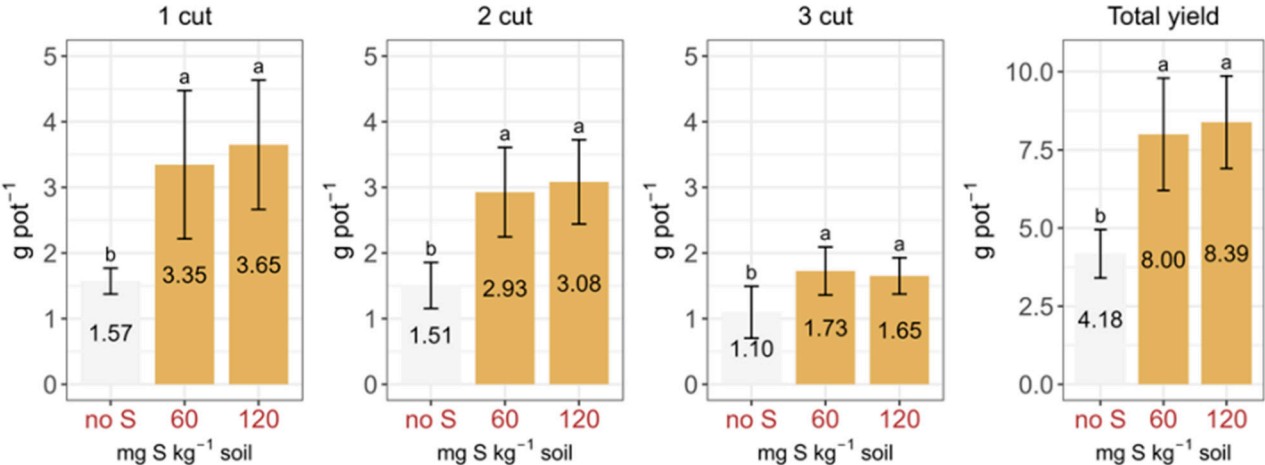

**Figure 5.** Dry mass of ryegrass in relation to S dose. Values labeled with the same letter are not significantly different ($p < 0.05$).

Degryse et al. [34] claim that both elemental sulfur and $S^0$- enriched sulfate fertilizers sustain plant sulfur requirements over a longer period than $SO_4$-S alone. Of the three swaths made in our experiment, the last one was significantly smaller than the earlier ones. This may have been due to the depletion of mineral nutrients in the soil as a result of their intensive uptake by the fast-growing grass. In the early stages of growth, the mineral requirements of plants are very high. Taube et al. [35] also found a varying response of successive regrowths of ryegrass to sulfur and nitrogen fertilization, with the greatest effect in the first regrowth. The results suggest that improved sulfur availability as a result of fertilization increases plant vigor, improves plant metabolism and photosynthetic activity, and ultimately results in better growth, higher stress tolerance, and improved nitrogen use efficiency. Our results are consistent with other studies that report a significant yield response of different crops to sulfur fertilization [25,27,34,36]. However, there are also reports indicating that plant growth response to sulfur addition is not always so positive and unequivocal [17,37]. Under field conditions, the response of plants to sulfur application is complex (among other immobilization, mineralization, leaching processes may have great impact) and, despite numerous studies, the exact requirements for sulfur fertilization are not known [34].

### 3.2. Nitrogen Content and Uptake

Nitrogen is one of the most important yield forming factors in agriculture. Literature data indicate that plant nitrogen content is significantly modified by the plant-available sulfur in the soil [27,37]. The results showed that the nitrogen content varied from one swath to the next, with the smallest differences (mostly statistically insignificant) occurring in the first swath (Table 2).

Under all applied treatments (form and dose of sulfur), the N concentration in the plants from the first swath ranged from 40.1 to 44.7 g $kg^{-1}$ dry matter. Plants from the first and second swaths contained significantly more nitrogen compared to the third swath. In the latter, under sulfur fertilization conditions, N concentration ranged from 19 g $kg^{-1}$ dry matter (for 120 mg $S^0$ + $K_2SO_4$) to 36.8 g $kg^{-1}$ dry matter (for 120 mg $S^0$) and the response to sulfur fertilization was similar for both sulfur doses. In plants not fertilized with sulfur, N concentrations were 42, 55.3, and 45.9 g $kg^{-1}$ dry mass in the first, second, and third swaths, respectively. Thus, nitrogen concentration in plants grown without sulfur addition was significantly higher than in plants fertilized with sulfur. The observed decrease was probably due to a dilution effect caused by the strong stimulation of ryegrass growth by sulfur fertilizers. As previously mentioned, the application of sulfur resulted in a significant increase in fresh and dry mass of ryegrass shoots (Figures 1 and 2). A decrease in the N content under sulfur fertilization was also observed in other plants [38]. Under

$S^0$ treatment, the average N content of the three swaths was similar to that recorded in the unfertilized plants and not significantly different from the sulfate-fertilized plants.

**Table 2.** Nitrogen content in perennial ryegrass.

| Treatments | N Content (g kg$^{-1}$ Dry Mass) | | | | | | | |
|---|---|---|---|---|---|---|---|---|
| | 1 Cut | | 2 Cut | | 3 Cut | | Mean in Cuts | |
| Sulfur dose 60 mg kg$^{-1}$ soil | | | | | | | | |
| no S | 42.0 | *ab* | 55.3 | *a* | 45.9 | *a* | 47.7 | *a* |
| $S^0$ | 42.5 | *ab* | 54.8 | *a* | 34.5 | *b* | 43.9 | *ab* |
| $S^0 + K_2SO_4$ | 43.1 | *ab* | 39.1 | *d* | 19.8 | *d* | 34.0 | *e* |
| $K_2SO_4$ | 42.4 | *ab* | 50.1 | *abc* | 29.8 | *b* | 40.8 | *bc* |
| $S^0 + MgSO_4$ | 40.1 | *b* | 45.3 | *bcd* | 20.2 | *cd* | 35.2 | *de* |
| $MgSO_4$ | 43.9 | *ab* | 53.3 | *a* | 29.0 | *bc* | 42.1 | *bc* |
| $S^0 + (NH_4)_2SO_4$ | 45.9 | *a* | 52.0 | *ab* | 29.5 | *b* | 42.5 | *bc* |
| $(NH_4)_2SO_4$ | 42.8 | *ab* | 43.5 | *cd* | 30.4 | *b* | 38.9 | *cd* |
| *Comparison of sulfur forms for dose 60 mg kg$^{-1}$ soil* | | | | | | | | |
| no S | 42.0 | *a* | 55.3 | *a* | 45.9 | *a* | 47.7 | *a* |
| $S^0$ | 42.5 | *a* | 54.8 | *a* | 34.5 | *b* | 43.9 | *ab* |
| $S^0 + SO_4$ | 43.0 | *a* | 45.5 | *b* | 23.2 | *c* | 37.2 | *c* |
| $SO_4$ | 43.0 | *a* | 49.0 | *ab* | 29.7 | *b* | 40.6 | *b* |
| Sulfur dose 120 mg kg$^{-1}$ soil | | | | | | | | |
| no S | 42.0 | *a* | 55.3 | *a* | 45.9 | *a* | 47.7 | *a* |
| $S^0$ | 41.5 | *a* | 50.6 | *a* | 36.8 | *b* | 43.0 | *ab* |
| $S^0 + K_2SO_4$ | 40.9 | *a* | 33.3 | *c* | 19.0 | *d* | 31.1 | *c* |
| $K_2SO_4$ | 42.5 | *a* | 50.6 | *a* | 28.2 | *c* | 40.4 | *b* |
| $S^0 + MgSO_4$ | 40.6 | *a* | 37.5 | *bc* | 21.7 | *cd* | 33.3 | *c* |
| $MgSO_4$ | 43.3 | *a* | 46.8 | *ab* | 28.2 | *c* | 39.4 | *b* |
| $S^0 + (NH_4)_2SO_4$ | 44.7 | *a* | 50.1 | *a* | 36.4 | *b* | 43.7 | *ab* |
| $(NH_4)_2SO_4$ | 44.5 | *a* | 48.1 | *a* | 29.5 | *bc* | 40.7 | *b* |
| *Comparison of sulfur forms for dose 120 mg kg$^{-1}$ soil* | | | | | | | | |
| no S | 42.0 | *a* | 55.3 | *a* | 45.9 | *a* | 47.7 | *a* |
| $S^0$ | 41.5 | *a* | 50.6 | *a* | 36.8 | *ab* | 43.0 | *ab* |
| $S^o + SO_4$ | 42.1 | *a* | 40.3 | *b* | 25.7 | *c* | 36.0 | *c* |
| $SO_4$ | 43.4 | *a* | 48.5 | *a* | 28.6 | *bc* | 40.2 | *b* |
| Comparison of sulfur doses: 60 and 120 mg kg$^{-1}$ soil | | | | | | | | |
| no S | 42.0 | *a* | 55.3 | *a* | 45.9 | *a* | 47.7 | *a* |
| 60 mg kg$^{-1}$ soil | 43.0 | *a* | 48.3 | *ab* | 27.6 | *b* | 39.6 | *b* |
| 120 mg kg$^{-1}$ soil | 42.6 | *a* | 45.3 | *b* | 28.5 | *b* | 38.8 | *b* |

Values labeled with the same letter are not significantly different ($p < 0.05$).

The uptake of nitrogen by plants from individual swaths showed a similar pattern to that observed in nitrogen concentration (Table 3). The lowest nitrogen uptake was recorded in the third swath, and this amount was several times lower than in the first swath.

This downward effect was probably due to the depletion of readily available nitrogen in the soil. The highest total uptake (sum of three swaths) was recorded with $(NH_4)_2SO_4$ application (both S doses) and the lowest with no sulfur fertilization. Sulfur fertilization markedly increased nitrogen uptake by ryegrass plants, which in turn contributed to a significant stimulation of plant growth. In general, the effect of both sulfur doses was similar, with the exception of the $S^0$ fertilization where the total nitrogen uptake was significantly greater under the higher S dose, and counted 303 mg pot$^{-1}$ against 245 for the lower dose. Reports on the interaction between N and S suggest that they affect each other, from the soil uptake, transport in the plant, and assimilation in cells [14]. Sulfur-deficient crops utilize nitrogen inefficiently, which results in increased losses of nitrogen to the environment [39,40].

**Table 3.** Nitrogen uptake of perennial ryegrass.

| Treatments | N Uptake (mg pot$^{-1}$) | | | | | | | |
|---|---|---|---|---|---|---|---|---|
| | 1 Cut | | 2 Cut | | 3 Cut | | Total Uptake | |
| Sulfur dose 60 mg kg$^{-1}$ soil | | | | | | | | |
| no S | 66.0 | *c* | 82.8 | *d* | 50.4 | *a* | 199 | *d* |
| S$^0$ | 94.6 | *c* | 113 | *bcd* | 44.2 | *a* | 245 | *cd* |
| S$^0$ + K$_2$SO$_4$ | 189 | *ab* | 138 | *abc* | 32.5 | *a* | 325 | *b* |
| K$_2$SO$_4$ | 88.6 | *c* | 110 | *cd* | 53.3 | *a* | 248 | *cd* |
| S$^0$ + MgSO$_4$ | 160 | *b* | 155 | *a* | 36.6 | *a* | 327 | *ab* |
| MgSO$_4$ | 174 | *ab* | 147 | *abc* | 51.1 | *a* | 356 | *ab* |
| S$^0$ + (NH$_4$)$_2$SO$_4$ | 97.3 | *c* | 151 | *ab* | 56.5 | *a* | 296 | *bc* |
| (NH$_4$)$_2$SO$_4$ | 199 | *a* | 156 | *a* | 54.3 | *a* | 389 | *a* |
| *Comparison of sulfur forms for dose 60 mg kg$^{-1}$ soil* | | | | | | | | |
| no S | 66.0 | *c* | 82.8 | *c* | 50.4 | *a* | 199 | *b* |
| S$^0$ | 94.6 | *bc* | 113 | *bc* | 44.2 | *a* | 252 | *b* |
| S$^0$ + SO$_4$ | 149 | *ab* | 148 | *a* | 41.9 | *a* | 339 | *a* |
| SO$_4$ | 154 | *a* | 138 | *ab* | 52.9 | *a* | 344 | *a* |
| Sulfur dose 120 mg kg$^{-1}$ soil | | | | | | | | |
| no S | 66.0 | *d* | 82.8 | *c* | 50.4 | *ab* | 199 | *d* |
| S$^0$ | 107 | *c* | 157 | *ab* | 51.3 | *ab* | 303 | *bc* |
| S$^0$ + K$_2$SO$_4$ | 185 | *ab* | 121 | *abc* | 33.6 | *b* | 308 | *abc* |
| K$_2$SO$_4$ | 178 | *ab* | 154 | *ab* | 44.1 | *b* | 355 | *ab* |
| S$^0$ + MgSO$_4$ | 158 | *b* | 125 | *abc* | 37.5 | *b* | 299 | *bc* |
| MgSO$_4$ | 189 | *a* | 126 | *abc* | 46.2 | *ab* | 342 | *ab* |
| S$^0$ + (NH$_4$)$_2$SO$_4$ | 86.8 | *cd* | 113 | *bc* | 67.7 | *a* | 265 | *c* |
| (NH$_4$)$_2$SO$_4$ | 181 | *ab* | 164 | *a* | 45.6 | *ab* | 367 | *a* |
| *Comparison of sulfur forms for dose 120 mg kg$^{-1}$ soil* | | | | | | | | |
| no S | 66.0 | *c* | 82.8 | *c* | 50.4 | *a* | 199 | *c* |
| S$^0$ | 107 | *bc* | 157 | *a* | 51.3 | *a* | 315 | *b* |
| S$^0$ + SO$_4$ | 143 | *b* | 120 | *b* | 46.3 | *a* | 309 | *b* |
| SO$_4$ | 183 | *a* | 148 | *a* | 45.3 | *a* | 376 | *a* |
| Comparison of sulfur doses: 60 and 120 mg kg$^{-1}$ soil | | | | | | | | |
| no S | 66 | *b* | 83 | *b* | 50.4 | *a* | 199 | *b* |
| 60 mg kg$^{-1}$ soil | 143 | *a* | 139 | *a* | 46.9 | *a* | 329 | *a* |
| 120 mg kg$^{-1}$ soil | 155 | *a* | 137 | *a* | 46.6 | *a* | 339 | *a* |

Values labeled with the same letter are not significantly different ($p < 0.05$).

### 3.3. Sulfur Content and Uptake

Overall, as expected, sulfur fertilization resulted in a better supply of sulfur to the ryegrass, and in some cases, the effect was more pronounced with a higher dose of sulfur. This was particularly evident after the application of fertilizers containing only sulfates. Under these conditions, the average sulfur content of the three swaths was 5.66 and 7.0 g kg$^{-1}$ dry mass for the 60 and 120 mg S kg$^{-1}$ soil doses, respectively (Table 4).

The highest sulfur concentration (as an average of three swaths) was recorded for plants fertilized with potassium sulfate (120 mg kg$^{-1}$ soil) and this value (8.02 g kg$^{-1}$ dry mass) was 70% higher than that of unfertilized plants. Fertilizing the plants with the mixture of elemental sulfur with potassium or magnesium sulfate did not increase the sulfur content in ryegrass shoots. Some positive effects were found only in the first swath. However, these were sufficient amounts for proper growth and function. Our results also indicate that ryegrass is capable of maintaining relatively high sulfur concentrations in shoots (4.74 g kg$^{-1}$ dry mass) despite low soil content. The recorded concentration is within the range expected for most crops (0.2 to 0.5% dry matter) [9]. However, a negative consequence was a strong inhibition of plant growth compared to sulfur-fertilized plants. In terms of sulfur requirements, multiple-cut grasses and Brassica crops are more prone to sulfur deficiency than other plants [20]. The sulfur demand is dependent not only on the plant species but also on its developmental stage [41]. During vegetative growth, sulfur uptake is optimized for growth. All plants regulate sulfur uptake and are able to adjust to

very variable (low and high soil sulfur levels) and time-varying sulfur supply [42]. The uptake of sulfur by unfertilized plants remained at a similar level in the next three swaths (6.41, 7.27, and 5.71 mg pot$^{-1}$, respectively; Table 5). As expected, sulfur-fertilized plants took up significantly larger amounts of sulfur and, in general, the values were higher at the increased S dose. Analyzing the dynamics of sulfur uptake by S-fertilized plants, it can be stated that it was greatest during the first two periods of growth (Table 5).

**Table 4.** Sulfur content in perennial ryegrass.

| Treatments | S Content (g kg$^{-1}$ Dry Mass) in Perennial Ryegrass | | | | | | | |
|---|---|---|---|---|---|---|---|---|
| | **1 Cut** | | **2 Cut** | | **3 Cut** | | **Mean in Cuts** | |
| Sulfur dose 60 mg kg$^{-1}$ soil | | | | | | | | |
| no S | 4.09 | *b* | 4.84 | *d* | 5.28 | *bc* | 4.74 | *d* |
| S$^0$ | 4.88 | *a* | 5.62 | *bc* | 6.27 | *ab* | 5.59 | *abc* |
| S$^0$ + K$_2$SO$_4$ | 4.72 | *ab* | 5.40 | *bcd* | 4.16 | *cd* | 4.76 | *d* |
| K$_2$SO$_4$ | 4.94 | *a* | 6.01 | *ab* | 7.33 | *a* | 6.09 | *a* |
| S$^0$ + MgSO$_4$ | 4.79 | *a* | 5.58 | *bc* | 3.96 | *d* | 4.78 | *d* |
| MgSO$_4$ | 4.55 | *ab* | 5.40 | *bcd* | 5.45 | *b* | 5.13 | *cd* |
| S$^0$ + (NH$_4$)$_2$SO$_4$ | 4.93 | *a* | 5.25 | *cd* | 5.86 | *b* | 5.35 | *bc* |
| (NH$_4$)$_2$SO$_4$ | 4.93 | *a* | 6.30 | *a* | 6.04 | *b* | 5.76 | *ab* |
| *Comparison of sulfur forms for dose 60 mg kg$^{-1}$ soil* | | | | | | | | |
| no S | 4.09 | *b* | 4.84 | *c* | 5.28 | *ab* | 4.74 | *b* |
| S$^0$ | 4.88 | *a* | 5.62 | *ab* | 6.27 | *a* | 5.59 | *a* |
| S$^0$ + SO$_4$ | 4.81 | *a* | 5.41 | *b* | 4.66 | *b* | 4.96 | *b* |
| SO$_4$ | 4.80 | *a* | 5.90 | *a* | 6.27 | *a* | 5.66 | *a* |
| Sulfur dose 120 mg kg$^{-1}$ soil | | | | | | | | |
| no S | 4.09 | *e* | 4.84 | *cd* | 5.28 | *de* | 4.74 | *d* |
| S$^0$ | 5.37 | *c* | 5.53 | *bc* | 6.77 | *c* | 5.89 | *c* |
| S$^o$ + K$_2$SO$_4$ | 4.96 | *cd* | 4.50 | *d* | 4.71 | *e* | 4.72 | *d* |
| K$_2$SO$_4$ | 6.88 | *a* | 6.00 | *b* | 11.2 | *a* | 8.02 | *a* |
| S$^0$ + MgSO$_4$ | 4.78 | *d* | 5.34 | *bc* | 4.43 | *e* | 4.85 | *d* |
| MgSO$_4$ | 5.15 | *cd* | 5.77 | *b* | 5.63 | *cde* | 5.52 | *c* |
| S$^0$ + (NH$_4$)$_2$SO$_4$ | 5.48 | *c* | 5.30 | *bcd* | 6.06 | *cd* | 5.61 | *c* |
| (NH$_4$)$_2$SO$_4$ | 6.21 | *b* | 7.58 | *a* | 8.56 | *b* | 7.45 | *b* |
| *Comparison of sulfur forms for dose 120 mg kg$^{-1}$ soil* | | | | | | | | |
| no S | 4.09 | *c* | 4.84 | *b* | 5.28 | *b* | 4.74 | *b* |
| S$^0$ | 5.37 | *ab* | 5.53 | *ab* | 6.77 | *ab* | 5.89 | *b* |
| S$^0$ + SO$_4$ | 5.07 | *b* | 5.04 | *b* | 5.07 | *b* | 5.06 | *b* |
| SO$_4$ | 6.08 | *a* | 6.45 | *a* | 8.46 | *a* | 7.00 | *a* |
| Comparison of Sulfur doses: 60 and 120 mg kg$^{-1}$ soil | | | | | | | | |
| no S | 4.09 | *c* | 4.84 | *b* | 5.28 | *b* | 4.74 | *b* |
| 60 mg kg$^{-1}$ soil | 4.82 | *b* | 5.65 | *ab* | 5.58 | *b* | 5.35 | *b* |
| 120 mg kg$^{-1}$ soil | 5.55 | *a* | 5.72 | *a* | 6.76 | *a* | 6.01 | *a* |

Values labeled with the same letter are not significantly different ($p < 0.05$).

For sulfur fertilization at 60 mg S kg$^{-1}$ soil, the highest total uptake (56.4 mg pot$^{-1}$, almost 3 times higher compared to the unfertilized plants) was observed in plants fertilized with ammonium sulfate. In plants exposed to 120 mg S, the highest sulfur uptake was found in the presence of potassium sulfate and ammonium sulfate.

The assimilation of sulfur and nitrogen are strongly linked, so a proper balance of these nutrients is very important. The N/S ratio is used to diagnose sulfur deficiency in plants [43–45]. Empirically, it has been established that for every 15 parts of N in a protein, there is 1 part of S. This means that the N/S ratio is fixed within a very narrow range of 15:1. Sulfur deficiency can limit protein synthesis, even if a large amount of N is available. This relationship has very important implications for human and animal nutrition.

Our results showed that for all fertilization variants, the N/S ratio was below the critical value of 15:1 (Table 6). For the averages of the three swaths, the ratio ranged from 5.82 to 10.3 and this highest value relates to non-fertilized plants. It may be considered that

the recorded N/S ratios (all below 15:1) were satisfactory. Similar trends were observed for both sulfur doses. The reduced N/S ratio indicates that the plants did not suffer from sulfur deficiency (based on the value of the N/S ratio), and that plants not fertilized with sulfur were able to take up sulfur despite the low soil content. However, the growth of plants was significantly affected.

**Table 5.** Sulfur uptake of perennial ryegrass.

| Treatments | S Uptake (mg pot$^{-1}$) | | | | | | | |
|---|---|---|---|---|---|---|---|---|
| | **1 Cut** | | **2 Cut** | | **3 Cut** | | **Total Uptake** | |
| Sulfur dose 60 mg kg$^{-1}$ soil | | | | | | | | |
| no S | 6.41 | *d* | 7.27 | *d* | 5.71 | *e* | 19.4 | *e* |
| $S^0$ | 10.9 | *c* | 11.6 | *cd* | 7.90 | *cde* | 30.4 | *d* |
| $S^0 + K_2SO_4$ | 20.7 | *ab* | 19.0 | *ab* | 6.85 | *de* | 46.5 | *b* |
| $K_2SO_4$ | 10.4 | *cd* | 13.5 | *c* | 12.9 | *a* | 36.7 | *cd* |
| $S^0 + MgSO_4$ | 19.2 | *ab* | 19.2 | *ab* | 7.21 | *de* | 45.5 | *b* |
| $MgSO_4$ | 17.9 | *b* | 14.8 | *bc* | 9.40 | *bcd* | 42.1 | *bc* |
| $S^0 + (NH_4)_2SO_4$ | 10.4 | *c* | 15.2 | *bc* | 11.3 | *ab* | 36.9 | *cd* |
| $(NH_4)_2SO_4$ | 22.9 | *a* | 22.6 | *a* | 10.9 | *abc* | 56.4 | *a* |
| *Comparison of sulfur forms for dose 60 mg kg$^{-1}$ soil* | | | | | | | | |
| no S | 6.41 | *b* | 7.27 | *b* | 5.71 | *b* | 19.4 | *b* |
| $S^0$ | 10.9 | *ab* | 11.6 | *b* | 7.90 | *b* | 30.4 | *b* |
| $S^0 + SO_4$ | 16.8 | *a* | 17.8 | *a* | 8.46 | *b* | 43.0 | *a* |
| $SO_4$ | 17.1 | *a* | 17.0 | *a* | 11.0 | *a* | 45.1 | *a* |
| Sulfur dose 120 mg kg$^{-1}$ soil | | | | | | | | |
| no S | 6.41 | *e* | 7.27 | *d* | 5.71 | *d* | 19.4 | *d* |
| $S^0$ | 13.8 | *d* | 17.0 | *bc* | 9.38 | *bcd* | 40.2 | *bc* |
| $S^0 + K_2SO_4$ | 22.4 | *bc* | 16.4 | *bc* | 8.49 | *cd* | 47.4 | *b* |
| $K_2SO_4$ | 28.7 | *a* | 18.5 | *b* | 17.5 | *a* | 64.8 | *a* |
| $S^0 + MgSO_4$ | 18.5 | *c* | 18.0 | *b* | 7.67 | *cd* | 44.2 | *b* |
| $MgSO_4$ | 22.5 | *bc* | 15.5 | *bc* | 9.21 | *bcd* | 47.3 | *b* |
| $S^0 + (NH_4)_2SO_4$ | 10.6 | *de* | 11.9 | *cd* | 11.3 | *bc* | 33.8 | *c* |
| $(NH_4)_2SO_4$ | 25.3 | *ab* | 25.7 | *a* | 13.4 | *ab* | 64.4 | *a* |
| *Comparison of sulfur forms for dose 120 mg kg$^{-1}$ soil* | | | | | | | | |
| no S | 6.41 | *c* | 7.27 | *c* | 5.71 | *b* | 19.4 | *c* |
| $S^0$ | 13.8 | *b* | 17.0 | *ab* | 9.38 | *ab* | 40.2 | *b* |
| $S^0 + SO_4$ | 17.2 | *b* | 15.4 | *b* | 9.15 | *b* | 41.8 | *b* |
| $SO_4$ | 25.5 | *a* | 19.9 | *a* | 13.4 | *a* | 58.8 | *a* |
| Comparison of sulfur doses: 60 and 120 mg kg$^{-1}$ soil | | | | | | | | |
| no S | 6.41 | *c* | 7.27 | *b* | 5.71 | *b* | 19.4 | *c* |
| 60 mg kg$^{-1}$ soil | 16.1 | *b* | 16.5 | *a* | 9.49 | *a* | 42.1 | *b* |
| 120 mg kg$^{-1}$ soil | 20.3 | *a* | 17.6 | *a* | 11.0 | *a* | 48.9 | *a* |

Values labeled with the same letter are not significantly different ($p < 0.05$).

The lowest N/S ratio value was found in the last swath, probably due to a reduced pool of plant-available nitrogen in the soil and its limited uptake (Table 5). In the third swath, N/S values ranged between 2.52 ($K_2SO_4$ in a higher dose) and 8.74 (unfertilized plants). Analyzing the results, it can be concluded that an N/S ratio below 9.0 provides an optimum yield of perennial ryegrass. Ref. [27], in determining the potential of sulfur fertilization to increase grassland yields and N use efficiency, found that the N/S ratio was variable during the growing season. Its values ranged from 5:1 to 20:1 depending on the harvest date and the applied sulfur fertilization. Ref. [46] claims that S supplementation is more necessary at higher N doses.

**Table 6.** N:S ratio in perennial ryegrass.

| Treatments | N:S in Perennial Ryegrass | | | | | | | |
|---|---|---|---|---|---|---|---|---|
| | **1 Cut** | | **2 Cut** | | **3 Cut** | | **Mean in Cuts** | |
| Sulfur dose 60 mg kg$^{-1}$ soil | | | | | | | | |
| no S | 10.3 | *a* | 11.4 | *a* | 8.74 | *a* | 10.3 | *a* |
| S$^0$ | 8.76 | *bc* | 9.78 | *b* | 5.52 | *b* | 8.30 | *bc* |
| S$^0$ + K$_2$SO$_4$ | 9.15 | *abc* | 7.26 | *c* | 4.81 | *b* | 7.73 | *cd* |
| K$_2$SO$_4$ | 8.62 | *bc* | 8.39 | *bc* | 4.10 | *b* | 6.92 | *d* |
| S$^0$ + MgSO$_4$ | 8.42 | *c* | 8.11 | *c* | 5.10 | *b* | 7.75 | *cd* |
| MgSO$_4$ | 9.66 | *ab* | 9.91 | *ab* | 5.42 | *b* | 8.81 | *b* |
| S$^0$ + (NH$_4$)$_2$SO$_4$ | 9.32 | *abc* | 9.92 | *ab* | 5.02 | *b* | 8.25 | *bc* |
| (NH$_4$)$_2$SO$_4$ | 8.69 | *bc* | 6.91 | *c* | 5.01 | *b* | 7.25 | *d* |
| *Comparison of sulfur forms for dose 60 mg kg$^{-1}$ soil* | | | | | | | | |
| no S | 10.3 | *a* | 11.4 | *a* | 8.74 | *a* | 10.3 | *a* |
| S$^0$ | 8.76 | *b* | 9.78 | *ab* | 5.52 | *b* | 8.30 | *b* |
| S$^0$ + SO$_4$ | 8.96 | *b* | 8.43 | *b* | 4.98 | *b* | 7.91 | *b* |
| SO$_4$ | 8.99 | *b* | 8.40 | *b* | 4.84 | *b* | 7.66 | *b* |
| Sulfur dose 120 mg kg$^{-1}$ soil | | | | | | | | |
| no S | 10.3 | *a* | 11.4 | *a* | 8.74 | *a* | 10.3 | *a* |
| S$^0$ | 7.75 | *bc* | 9.18 | *bc* | 5.49 | *bc* | 7.81 | *b* |
| S$^0$ + K$_2$SO$_4$ | 8.26 | *bc* | 7.38 | *cde* | 4.10 | *cd* | 7.18 | *b* |
| K$_2$SO$_4$ | 6.17 | *d* | 8.52 | *bcd* | 2.52 | *e* | 5.82 | *c* |
| S$^0$ + MgSO$_4$ | 8.53 | *b* | 6.99 | *de* | 4.97 | *bc* | 7.27 | *b* |
| MgSO$_4$ | 8.41 | *b* | 8.15 | *bcde* | 5.02 | *bc* | 7.66 | *b* |
| S$^0$ + (NH$_4$)$_2$SO$_4$ | 8.19 | *bc* | 9.47 | *b* | 6.00 | *b* | 7.91 | *b* |
| (NH$_4$)$_2$SO$_4$ | 7.19 | *cd* | 6.36 | *e* | 3.47 | *de* | 6.05 | *c* |
| *Comparison of sulfur forms for dose 120 mg kg$^{-1}$ soil* | | | | | | | | |
| no S | 10.3 | *a* | 11.4 | *a* | 8.74 | *a* | 10.3 | *a* |
| S$^0$ | 7.75 | *bc* | 9.18 | *b* | 5.49 | *b* | 7.81 | *b* |
| S$^0$ + SO$_4$ | 8.33 | *b* | 7.95 | *b* | 5.02 | *b* | 7.45 | *b* |
| SO$_4$ | 7.26 | *c* | 7.68 | *b* | 3.67 | *c* | 6.51 | *c* |
| Comparison of sulfur doses: 60 and 120 mg kg$^{-1}$ soil | | | | | | | | |
| no S$^0$ | 10.3 | *a* | 11.4 | *a* | 8.74 | *a* | 10.3 | *a* |
| 60 mg kg$^{-1}$ soil | 8.95 | *b* | 8.61 | *b* | 5.00 | *b* | 7.52 | *b* |
| 120 mg kg$^{-1}$ soil | 7.79 | *c* | 8.01 | *b* | 4.51 | *b* | 6.77 | *c* |

Values labeled with the same letter are not significantly different ($p < 0.05$).

### 3.4. Soil Properties after Experiment

Fertilization with sulfur in both elemental and sulfate forms significantly decreased the soil reaction as compared to the object where this nutrient was not applied (Table 7). It was also found that the size of the sulfur dose had a significant effect on the decrease in soil reaction. It is often emphasized in the literature that the application of sulfur can affect the acidity of soils, especially when applied at higher doses [47–50].

In our study, sulfur fertilization also contributed to a significant decrease in both soil carbon and nitrogen content. The greatest decrease in soil carbon and nitrogen content was observed when a higher dose of sulfur was applied in its elemental form. The availability of sulfur to plants is related to the transformation of organic matter in the soil, which mainly depends on soil microbial activity. In turn, soil microbial activity is dependent on temperature, moisture, pH, and substrate availability [51–53]. Ikoyi et al. [54] found that short-term sulfate fertilization promotes perennial ryegrass growth by outweighing negative feedback from some of the soil biota, while Magnucka et al. [55] found that the application of sulfur fertilizers with the mineral NPKMg promotes soil fertility due to aggregate stabilization and a decrease in water-soluble organic compounds. Nitrogen mineralization from soil organic matter can occur more rapidly compared to the release of sulfur [56,57].

**Table 7.** Physical-chemical soil properties after experiment.

| Treatments | pH | | $C_{organic}$ | | $N_{total}$ | | $S_{total}$ | | S-SO$_4$ | | S-SO$_4$/S$_{total}$ | |
|---|---|---|---|---|---|---|---|---|---|---|---|---|
| | 1M KCl dm$^{-3}$ | | (g kg$^{-1}$ Soil) | | | | (mg kg$^{-1}$ Soil) | | | | % | |
| colspan Sulfur dose 60 mg kg$^{-1}$ soil | | | | | | | | | | | | |
| no S | 6.59 | a | 8.58 | a | 0.610 | a | 94 | e | 3.99 | f | 4.23 | c |
| S$^0$ | 6.47 | b | 6.83 | bc | 0.544 | abc | 106 | d | 12.7 | a | 11.9 | a |
| S$^0$ + K$_2$SO$_4$ | 6.48 | b | 7.27 | b | 0.522 | bcd | 136 | a | 10.4 | bc | 7.65 | b |
| K$_2$SO$_4$ | 6.43 | b | 6.86 | bc | 0.468 | d | 104 | d | 6.02 | ef | 6.35 | b |
| S$^0$ + MgSO$_4$ | 6.43 | b | 5.84 | c | 0.568 | ab | 126 | ab | 7.65 | de | 6.06 | bc |
| MgSO$_4$ | 6.48 | b | 7.83 | ab | 0.532 | bcd | 110 | cd | 7.06 | de | 6.43 | b |
| S$^0$ + NH$_4$)$_2$SO$_4$ | 6.38 | b | 6.96 | b | 0.496 | cd | 117 | bc | 11.7 | ab | 10.1 | a |
| (NH$_4$)$_2$SO$_4$ | 6.44 | b | 7.74 | ab | 0.554 | abc | 133 | a | 8.58 | cd | 6.47 | b |
| *Comparison of sulfur forms for dose 60 mg kg$^{-1}$ soil* | | | | | | | | | | | | |
| no S | 6.59 | a | 8.58 | a | 0.610 | a | 94 | c | 3.99 | d | 4.23 | d |
| S$^0$ | 6.47 | b | 6.83 | bc | 0.544 | ab | 106 | b | 12.7 | a | 11.9 | a |
| S$^0$ + SO$_4$ | 6.43 | b | 6.69 | c | 0.529 | b | 126 | a | 9.92 | b | 7.93 | b |
| SO$_4$ | 6.45 | b | 7.48 | b | 0.518 | b | 116 | ab | 7.22 | c | 6.42 | c |
| colspan Sulfur dose 120 mg kg$^{-1}$ soil | | | | | | | | | | | | |
| no S | 6.59 | a | 8.58 | a | 0.610 | a | 94 | e | 3.99 | c | 4.23 | e |
| S$^0$ | 6.37 | b | 5.39 | c | 0.466 | c | 147 | cd | 23.0 | a | 15.7 | a |
| S$^0$ + K$_2$SO$_4$ | 6.36 | b | 7.62 | ab | 0.518 | bc | 158 | b | 21.8 | a | 13.8 | ab |
| K$_2$SO$_4$ | 6.36 | b | 7.34 | b | 0.498 | bc | 144 | d | 13.5 | b | 9.37 | cd |
| S$^0$ + MgSO$_4$ | 6.36 | b | 6.84 | b | 0.536 | bc | 159 | ab | 19.7 | a | 12.4 | bc |
| MgSO$_4$ | 6.41 | b | 7.73 | ab | 0.526 | bc | 153 | bc | 10.7 | b | 7.00 | de |
| S$^0$ + (NH$_4$)$_2$SO$_4$ | 6.40 | b | 7.20 | b | 0.500 | bc | 146 | cd | 21.4 | a | 14.6 | ab |
| (NH$_4$)$_2$SO$_4$ | 6.37 | b | 7.57 | b | 0.556 | ab | 167 | a | 10.8 | b | 6.52 | de |
| *Comparison of sulfur forms for dose 120 mg kg$^{-1}$ soil* | | | | | | | | | | | | |
| no S | 6.59 | a | 8.58 | a | 0.610 | a | 94 | b | 3.99 | c | 4.23 | c |
| S$^0$ | 6.37 | b | 5.39 | c | 0.466 | c | 147 | a | 23.0 | a | 15.7 | a |
| S$^0$ + SO$_4$ | 6.37 | b | 7.22 | b | 0.518 | bc | 154 | a | 21.0 | a | 13.6 | a |
| SO$_4$ | 6.38 | b | 7.55 | b | 0.527 | b | 155 | a | 11.7 | b | 7.63 | b |
| colspan Comparison of sulfur doses: 60 and 120 mg kg$^{-1}$ soil | | | | | | | | | | | | |
| no S$^0$ | 6.59 | a | 8.58 | a | 0.610 | a | 94 | c | 3.99 | c | 4.23 | c |
| 60 mg kg$^{-1}$ soil | 6.44 | b | 7.05 | b | 0.526 | b | 119 | b | 9.15 | b | 7.78 | b |
| 120 mg kg$^{-1}$ soil | 6.38 | c | 7.10 | b | 0.514 | b | 153 | a | 17.3 | a | 11.3 | a |

Values labeled with the same letter are not significantly different ($p < 0.05$).

Under the influence of sulfur fertilization with all analyzed forms of sulfur, the content of total sulfur and sulfate sulfur in the soil increased significantly compared to the control object. Similar studies comparing sulfur fertilization in the sulfate and elementary form also found similar relationships in the content of this element in soil [49]. This may be due to the fact that sulfur applied in the sulfate form is directly available to plants and a significantly higher overall uptake of sulfur by ryegrass was observed in comparison with objects fertilized with the elementary form. This confirms that fertilization with elemental forms of sulfur has a greater subsequent effect of sulfur delivery from the soil compared to sulfate forms.

Maintaining the optimal soil abundance of available forms of sulfur can have both agronomic and environmental benefits [27].

Increasing doses of sulfur also had a significant effect on the content of this element in the soil. Both lower and higher doses of sulfur were found to have the highest sulfate content in the soil when fertilized with the elemental form of sulfur, indicating a slower release of the available sulfur form into the soil during plant growth. This was also confirmed by the highest proportion of the total sulfate form in the soil when fertilizing with elemental sulfur. The combined use of elemental sulfur and sulfate sulfur also significantly increased the sulfate sulfur content of the soil compared with the sulfate form of sulfur. It is safer to use the elemental form of sulfur at higher sulfur rates applied to the grassland due to sulfate leaching into the soil profile [58].

## 4. Conclusions

The results indicate that fertilizers that are a mixture of S⁰ and sulfate are as effective as those containing only sulfate in terms of stimulating perennial ryegrass growth. Both applied doses of sulfur were equally effective. Fertilization with all forms of sulfur increased the soil's content of this nutrient, making it possible to compensate for deficiencies of this element in the soil. Economic and environmental considerations indicate that the use of fertilizers that are a mixture of elemental and sulfate sulfur is a good approach in agriculture.

**Author Contributions:** Conceptualization, G.K., Ł.M. and E.S; methodology, G.K., A.K.-L. and E.S.; investigation, G.K. and E.S.; data curation—compiled and analyzed the results, G.K., E.S. and A.K.-L.; writing—original draft preparation, E.S. and G.K.; writing—review and editing, G.K., A.K.-L. and E.S. All authors have read and agreed to the published version of the manuscript.

**Funding:** These studies were co-financed by the National Center for Research and Development, Poland, under contract No. POIR.01.01.02-00-0145/16-00.

**Data Availability Statement:** The data presented in this manuscript are available from the authors upon reasonable request.

**Conflicts of Interest:** The authors declare no conflict of interest.

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
