# Peer review of "Perennial Ryegrass (Lolium perenne L.) Response to Different Forms of Sulfur Fertilizers"

_agriculture, doi:10.3390/agriculture13091773_

Round 1

Reviewer 1 Report

Dear authors,

You are presenting an interesting work but see some comments to improve the justification of some of your arguments and improve the presentation of the article. Particularly, it is concerning the way you presented and described the figures and tables. Only the most relevant data should be included in the main manuscript and all the other supporting data needs to be moved to the supplementary material or appendix. Please consider the following comments:

-          Line 12: “Correspondence” written twice.

-          Lines 29 to 36: It is not clear or very well justified the need for an investigation in reducing the use of sulfur fertilizers, from the point of view of depleting the resources of sulfur that the authors are offering in their manuscript. The statement of: “In recent years, there is observed a shortage of plant-available sulphur in soils almost all over the world” needs to be backed up with data available in the literature. The article that they are citing in the first paragraph of the introduction [1] (https://doi.org/10.1080/16000889.2017.1328945) describes the deposition of nitrogen and sulfur, as consequence of the excessive air contamination. In the next sentence, the authors subsequently claim that reducing the depositions of sulfur (due to cultivation of high-yielding crops that reduce the sulfurous air pollutants) is a bad thing, which is difficult to believe. Please, re-elaborate this paragraph. For instance, the next paragraph (lines 37 to 51) makes more sense since the authors describe the difficulties of making an efficient use of sulfur and the challenges to get this element absorbed by crops and preventing leaching. This is an argument that is more reliable than the affirmations made in the first paragraph. It is not necessary to completely discard the first paragraph of the introduction because is an original argument, but it needs to be better justified.

-          Line 46: What is the meaning of “sulfur is not mobile in the plants”? What about the other nutrients? Are they mobile? And how the fact that the sulfur is not mobile in the plants is connected to the second part of the sentence (i.e. sulfur should be available throughout the growing season)?

-          Line 52 to 61: These 2 paragraphs should be merged into a single paragraph.

-          Line 62: it is not clear what is the rationale for the hypothesis of using elemental sulfur. Is it because is less soluble than sulfate? Please state it clearer.

-          Figure 1: Please, remove the text “dose of sulphur – 60 mg/kg soil” above the graphs because this information is already described in the caption of the figure.

-          Figure 2: Please, remove the text “dose of sulphur – 120 mg/kg soil” above the graphs because this information is already described in the caption of the figure.

-          Line 152: Please, remove the term “clearly” to make the description of the findings more formal and objective.

-          Lines 152 to 154: Please, describe and discuss each figure separately. It is not convenient for the readers to have all figures in a row, without any text description.

-          Figure 1 to Figure 10: Please remove the redundant figures. Readers appreciate a concise article.

-          Line 353: “Krrr” needs to be corrected.

-          Line 375: Please, remove the term “clearly” to make more objective and formal the description of the investigation.

-          Lines 379 and 380: Please, rewrite the sentence: “It is worth noting that the lower dose of sulfur applied was as effective as the higher rate.”

-          Lines 381 to 385: This second paragraph of conclusions should be moved before the first paragraph. This was the same order in which you presented the article, initially talking about the sulfur deficiency and then about the use of elemental sulfur to maximize the use efficiency of this nutrient upon land application.

-          Line 12: “Correspondence” written twice.

-          Line 152: Please, remove the term “clearly” to make the description of the findings more formal and objective.

-          Line 353: “Krrr” needs to be corrected.

-          Line 375: Please, remove the term “clearly” to make more objective and formal the description of the investigation.

Author Response

Dear Reviewer 1,

Thank you very much for your insightful comments and valuable suggestions which will help to improve our manuscript. All issues raised by the reviewer have been clarified and discussed as follows:

  1. Line 12: “Correspondence” written twice.

Removed

  1. Lines 29 – 36.

Added literature items indicating a decrease in sulphur content in soils.

“Sulphur (S), an macronutrient found in plants in the smallest amounts relative to other essential macronutrients, is currently recognised as one of the most important yield- forming elements. In recent years, there is observed a shortage of plant-available sulphur in soils almost all over the world”.

To make next sentence more clearer we changed it slightly.

We removed this sentence:

“This adverse phenomenon results from a cultivation of  high-yielding crops and a drastic reduction in sulphurous air pollutants and consequently reduced S deposition in the soil.”

Added:

“This adverse phenomenon is due to a drastic reduction in air pollution by sulphur, and consequently a reduction in its deposition in the soil. High-yielding crops grown today take up large amounts of sulphur from the soil and exacerbate this problem.”

  1. Line 46: What is the meaning of “sulfur is not mobile in the plants”? What about the other nutrients? Are they mobile? And how the fact that the sulfur is not mobile in the plants is connected to the second part of the sentence (i.e. sulfur should be available throughout the growing season)?

This sentence is modified and expanded:

“On the other hand, sulphur is classified as an immobile mineral element in the plant that is not readily remobilized to younger leaves during deficiency. Its deficiency at any stage of plant growth negatively affects plant metabolism and growth, and ultimately results in reduced yields.”

  1. Line 52 to 61: These 2 paragraphs should be merged into a single paragraph

Paragraphs merged

  1. Line 62: it is not clear what is the rationale for the hypothesis of using elemental sulfur. Is it because is less soluble than sulfate? Please state it clearer.

Added a sentence of clarification:

“Elemental sulphur is chemically inert, difficult to leaching from the soil compared to the anionic sulphate form, and longer available in the soil”.

  1. Figure 1: Please, remove the text “dose of sulphur – 60 mg/kg soil” above the graphs because this information is already described in the caption of the figure.
  2. Figure 2: Please, remove the text “dose of sulphur – 120 mg/kg soil” above the graphs because this information is already described in the caption of the figure.

Thank you for the suggestion, but in our opinion, leaving these inscriptions on the drawings makes them more legible.

  1. Line 152: Please, remove the term “clearly” to make the description of the findings more formal and objective.

Removed

  1. Lines 152 to 154: Please, describe and discuss each figure separately. It is not convenient for the readers to have all figures in a row, without any text description.

Links to drawings have been completed and text has been slightly reformatted to make it more readable.

  1. Figure 1 to Figure 10: Please remove the redundant figures. Readers appreciate a concise article.

In our opinion, the presented figures 1 -10 visualize well the obtained results and facilitate their interpretation.

  1. Line 353: “Krrr” needs to be corrected.

The sentence has been reworded

  1. Line 375: Please, remove the term “clearly” to make more objective and formal the description of the investigation.

Removed

  1. Lines 379 and 380: Please, rewrite the sentence: “It is worth noting that the lower dose of sulfur applied was as effective as the higher rate.”

The sentence has been reworded

  1. Lines 381 to 385: This second paragraph of conclusions should be moved before the first paragraph. This was the same order in which you presented the article, initially talking about the sulfur deficiency and then about the use of elemental sulfur to maximize the use efficiency of this nutrient upon land application.

The conclusions have been rewritten.

“The results indicate that fertilizers that are a mixture of S0 and sulphate are as effective as those containing only sulphate in terms of stimulating perennial ryegrass growth. Both applied doses of sulphur were equally effective. Fertilization all forms of sulphur increased the soil's content of this nutrient, making it possible to compensate for deficiencies of this element in the soil. Economic and environmental considerations indicate that the use of fertilizers that are a mixture of elemental and sulphate sulphur is a good approach in agriculture.”

Reviewer 2 Report

The article entitled "Perennial ryegrass (Lolium perenne L.) response to different forms of sulphur fertilizers " is interesting and falls within the scope of the journal. Overall the manuscript is well structures and the findings are worth to the journal's audience, however there are some points that need further clarification:

1. Line 13-25: Please add some data to the abstract.

2. Please briefly introduce the perennial ryegrass in the “Introduction” chapter.

3. The statement in Line29-33, 52-55, and 76-79 should be supplemented by reference.

4. Line 41: What are the units of 0.2 and 0.5?

5. Line 29-81: The relationship between research background and research content is not close. It is necessary to summarize the research progress related to this study, so as to put forward the purpose and significance of this study.

6. Line85-90: Please supplement the source of data (Table 1).

7. How to control the water-holding capacity was kept at 60 %, weighing method?

8. Line131-132: Why choose to dry at 105℃ to a constant weight?

9. Line132-133: The methods in Line 132-133 should be supplemented by reference.

10. Line144-149: Those sentences should be presented in the “Introduction” and “Materials and Methods” chapter.

11. Line211: please delete “also”

12. Line 230: The statement in Line 230 should be supplemented by reference. Please used the reference: https://doi.org/10.3390/agronomy13071886

13. Line289: Please delete “;”

14. The statement in Line 289-292 should be supplemented by reference.

Language may be improved 

Author Response

Dear Reviewer 2,

Thank you very much for your insightful comments and valuable suggestions which will help to improve our manuscript. All issues raised by the reviewer have been clarified and discussed as follows:

  1. Line 13-25: Please add some data to the abstract.

The early version of the abstract contained a broader description of the results but due to publishing limitations the abstract had to be shortened

  1. Please briefly introduce the perennial ryegrass in the “Introduction” chapter.

Added to the introduction

Perennial ryegrass - a species of the grass family - grows wild almost throughout Europe, northern Africa, the temperate zone of Asia, artificially introduced in North America and Australia. An excellent pasture grass that forms the basis of productive pastures for cattle. It can also be used to stabilize soil and prevent soil erosion, as well as to create hardy turf for lawns and golf courses. It has produced many varieties. Ryegrass is highly productive plant and can be harvested multiple times during the growing season.

There are a relatively small number of studies that examine the sulphur fertilization needs of grasslands, especially in the temperate zone and over the past decade under conditions of very limited sulphur input from the atmosphere. Aspel et al. (Aspel et al., 2022) in lysimeter experiments on perennial ryegrass showed that sulphur fertilization increased crop yields, nitrogen recovery from fertilizers and significantly reduced nitrate leaching.

  1. The statement in Line29-33, 52-55, and 76-79 should be supplemented by reference.

Reference items added

  1. Line 41: What are the units of 0.2 and 0.5?

Unit added (%)

  1. Line 29-81: The relationship between research background and research content is not close. It is necessary to summarize the research progress related to this study, so as to put forward the purpose and significance of this study.

The introduction was supplemented

  1. Line85-90: Please supplement the source of data (Table 1).

Reference items added

  1. How to control the water-holding capacity was kept at 60 %, weighing method?

The full water capacity of the soil was determined, and then 60% was calculated. Watering to field capacity was done by weighing the pots and adding the appropriate amount of water.

  1. Line131-132: Why choose to dry at 105℃ to a constant weight?

Paragraph corrected

Plants were collected after each grass cutting and the fresh mass of ryegrass was determined. Then plants were dried at 105°C for one hour (to kill plant tissues and avoid dry matter loss through respiration) and then at 60°C to constant weight.

  1. Line132-133: The methods in Line 132-133 should be supplemented by reference.

Reference items added

  1. Line144-149: Those sentences should be presented in the “Introduction” and “Materials and Methods” chapter.

This part of the text has been deleted

  1. Line211: please delete “also”

Word has been deleted

  1. Line 230: The statement in Line 230 should be supplemented by reference. Please used the reference: https://doi.org/10.3390/agronomy13071886

A related publication was added

  1. Line289: Please delete “;”

Deleted

  1. The statement in Line 289-292 should be supplemented by reference.

Reference item added

Reviewer 3 Report

I appreciate the efforts of authors in bringing the manuscript technically sound with the substantial outcomes. However, I have few suggestions for the authors to further improve the overall quality of the manuscript. 

Introduction: Include the cultivation and production, role of sulphur and its importance in Ryegrass. 

Results and discussion section has to be improved with recent supporting literatures. 

Conclusion has to be rewritten considering the substantial outcomes and highlighting the major findings with future scope and recommendations. 

Minor english correction is required. 

Author Response

Dear Reviewer 3,

Thank you very much for your insightful comments and valuable suggestions which will help to improve our manuscript. All issues raised by the reviewer have been clarified and discussed as follows:

  1. Introduction: Include the cultivation and production, role of sulphur and its importance in Ryegrass.

The introduction has been expanded to include issues related to Ryegrass

  1. Results and discussion section has to be improved with recent supporting literatures.

Added to the results and discussion is the latest literature position

  1. Conclusion has to be rewritten considering the substantial out comes and highlighting the major findings with future scope and recommendations.

The conclusions have been rewritten

“The results indicate that fertilizers that are a mixture of S0 and sulphate are as effective as those containing only sulphate in terms of stimulating perennial ryegrass growth. Both applied doses of sulphur were equally effective. Fertilization all forms of sulphur increased the soil's content of this nutrient, making it possible to compensate for deficiencies of this element in the soil. Economic and environmental considerations indicate that the use of fertilizers that are a mixture of elemental and sulphate sulphur is a good approach in agriculture.”

Round 2

Reviewer 1 Report

Dear authors,

Please consider the following comments:

-          Lines 32 to 34: The statement “This adverse phenomenon is due to a drastic reduction in air pollution by sulphur, and consequently a reduction in its deposition in the soil.” would be justified with at least 1 literature reference.

-          Lines 252 to 263: Figures should appear in the manuscript in the same order as they are cited in the text. Please, improve this part, which is a major problem in the current presentation of the manuscript.

Please make sure that you are using either British or American English. For instance, fertilizer vs fertilizer or sulphur vs sulfur. Please proofread the whole text accordingly.

Author Response

Dear Reviewer 1, (Round2)

Thank you very much for your insightful comments and valuable suggestions which will help to improve our manuscript. All issues raised by the reviewer have been clarified and discussed as follows:

  • Lines 32 to 34: The statement “This adverse phenomenon is due to a drastic reduction in air pollution by sulphur, and consequently a reduction in its deposition in the soil.” would be justified with at least 1 literature reference.

Reference items added

  • Lines 252 to 263: Figures should appear in the manuscript in the same order as they are cited in the text. Please, improve this part, which is a major problem in the current presentation of the manuscript.

This part of the manuscript has been completely revised in accordance with suggestions, the number of figures has been reduced and their references in the text have been placed accordingly.

  • Please make sure that you are using either British or American English. For instance, fertilizer vs fertilizer or sulphur vs sulfur. Please proofread the hole text accordingly.

Correction of words was made

Reviewer 2 Report

Thank you for your great efforts in improving the manuscript. It is improved dramatically. Now, the paper should be accepted for the publication.

Minor editing of English language required

Author Response

Dear Reviewer 2, (Round2)

Thank you very much for your insightful comments and valuable suggestions which will help to improve our manuscript. All issues raised by the reviewer have been clarified and discussed as follows:

  • Minor editing of English language required

The English language has been corrected